# Uncertainty Quantification in Neural Differential Equations

**Olga Graf**[*]
Technical University of Munich
graf@ma.tum.de

**Pablo Flores**[*]
Pontificia Universidad Católica de Chile
ptflores1@uc.cl

**Pavlos Protopapas**
Harvard University
pavlos@seas.harvard.edu

**Karim Pichara**
Pontificia Universidad Católica de Chile
kpb@ing.puc.cl

## Abstract

Uncertainty quantification (UQ) helps to make trustworthy predictions based on collected observations and uncertain domain knowledge. With increased usage of deep learning in various applications, the need for efficient UQ methods that can make deep models more reliable has increased as well. Among applications that can benefit from effective handling of uncertainty are the deep learning based differential equation (DE) solvers. We adapt several state-of-the-art UQ methods to get the predictive uncertainty for DE solutions and show the results on four different DE types.

## 1 Introduction

Driven by the growing popularity of deep learning, several areas of research have obtained state-of-the-art performances with deep neural networks (NNs). Among other applications, deep NNs have been applied for solving differential equations (DEs) [10] — a fundamental tool for mathematical modeling in engineering, finance, and the natural sciences. Deep learning based solutions of DEs have recently appeared in, e.g., [20], [18], [19], [22], [9], [11], [17], [5], [21], [8]. Typically, the NN itself approximates the solution of a DE. Thanks to that, parallelization is natural and, in contrast to classical numerical methods, the solution at any time can be computed without the burden of having to compute all previous time steps. Furthermore, NNs are continuous and differentiable.

Until recently, the focus of deep learning was on achieving better accuracy in the NN predictions, but now it is increasingly being shifted to measuring the prediction's uncertainty, especially if the task at hand is safety critical. Uncertainty quantification (UQ) has been considered for deep models in computer vision, medical image analysis, bionformatics, etc [1]. Likewise, UQ is important for deep models that solve DEs. The uncertainty here stems from the fact that we cannot train a NN on an infinite time and/or space domain. Therefore, we seek to estimate the solution's uncertainty in the regions where the model was not trained. Moreover, another source of uncertainty comes from the model's limitations such as its architecture.

To the best of our knowledge, this is the first work to discuss UQ for deep models that solve DEs. Contrary to common deep learning setup, we solve DEs without any observed data, relying only on the samples of time and/or space and on the mathematical statement that relates functions and their derivatives. This makes the application of existing UQ methods not so straightforward. In this paper, we make the following contributions:

---

[*]These authors have contributed equally to this work.

35th Conference on Neural Information Processing Systems (NeurIPS 2021), Sydney, Australia.

1. We propose an adaptation of the four state-of-the-art UQ methods in deep learning — *Bayes By Backprop* [3], *Flipout* [24], *Neural Linear Model* [23, 15], and *Deep Evidential Regression* [2, 13] — to the case of solving DEs.

2. We test the above-mentioned methods on four different DE types: linear ordinary DE (ODE), non-linear ODE, system of non-linear ODEs, and partial DE (PDE).

## 2 Preliminaries

### 2.1 Solving differential equations with neural networks

A DE can be expressed as $\mathcal{L}u - f = 0$, where $\mathcal{L}$ is the differential operator, $u(\mathbf{x})$ is the solution that we wish to find on some (possibly multidimensional) domain $\mathbf{x}$, and $f$ is a known forcing function. We denote the NN approximation of the true solution by $u_N$. To solve the DE, we minimize the square loss of the residual function $\mathcal{R}(u_N) := \mathcal{L}u_N - f$, i.e., the optimization objective is

$$\min_{\mathbf{w}}(\mathcal{R}^2(u_N)), \tag{1}$$

where $\mathbf{w}$ are the NN parameters. It is also necessary to inform the NN about any initial and/or boundary conditions, $u_c = u(\mathbf{x}_c)$. One can achieve that in a straightforward way by adding a penalizing term to the loss function. However, the exact satisfaction of initial/boundary conditions is not possible in this case, causing problems in case of high sensitivity to initial conditions, and also yielding unnecessary local uncertainty at $\mathbf{x}_c$. Therefore, we employ an alternative approach and consider a transformation of $u_N$ which enforces the initial/boundary conditions and satisfies them by construction. E.g., in one-dimensional case, given an initial condition $u_0 = u(t_0)$, we consider a transformation $\tilde{u}_N(t) = u_0 + (1 - e^{-(t-t_0)})u_N(t)$. In general, the transformation has the form $\tilde{u}_N(\mathbf{x}) = A(\mathbf{x}, \mathbf{x}_c, u_c) + B(\mathbf{x}, \mathbf{x}_c)u_N(\mathbf{x})$. Hereinafter, $\tilde{u}_N$ will denote the enforced solution. Accordingly, we replace $u_N$ by $\tilde{u}_N$ in the optimization objective (1). Besides the advantage of satisfying the initial/boundary conditions exactly, the latter approach can also reduce the effort required during training [12].

### 2.2 Uncertainty quantification under Bayesian framework

While classical learning considers *deterministic model parameters* $\theta$, the Bayesian framework introduces uncertainty by considering a *posterior distribution over the model parameters*, $p(\theta|\mathcal{D})$, obtained after observing some data $\mathcal{D}$. The posterior distribution is given by Bayes' theorem, $p(\theta|\mathcal{D}) = p(\mathcal{D}|\theta) \cdot p(\theta) / p(\mathcal{D})$, where $p(\mathcal{D}|\theta)$ is the likelihood, $p(\theta)$ is the prior distribution over the parameters, and $p(\mathcal{D})$ is the evidence. The predictions $y$ at a new test point $\mathbf{x}$ are given by the posterior predictive distribution,

$$p(y|\mathbf{x}, \mathcal{D}) = \int p(y|\mathbf{x}, \theta) \cdot p(\theta|\mathcal{D})d\theta. \tag{2}$$

For probabilistic deep models, there are two main strategies of estimating (2).

**Inference through the posterior distribution of model parameters**. As stated in (2), the posterior predictive is obtained by averaging over the posterior uncertainty in the model parameters. Thus, we can start with estimating the posterior distribution of the NN weights.

In this case, well-suited is Bayesian NN [14] which places a prior distribution on all the weights (and biases) $\mathbf{w}$. Since an analytical solution for the posterior is intractable for Bayesian NNs, we have to use numerical approximation methods such as MCMC or variational methods. Despite the need for sampling in both cases, variational methods are computationally less expensive for high-dimensional parameter spaces and also provide an analytical approximation.

*Bayes By Backprop* (*BBB*) is a variational, backpropagation-compatible method for training a Bayesian NN. Its optimization objective seeks to minimize the Kullback-Leibler divergence between the true posterior and the variational posterior which is re-parametrized as $\mathcal{N}(\mu, \sigma = \log(1 + \exp(\rho)))$ to allow for backpropagation. At each optimization step, weights $\mathbf{w} = \mu + \sigma \circ \epsilon$, where $\epsilon \sim \mathcal{N}(0, I)$ and $\circ$ is pointwise multiplication, are obtained by sampling from the variational posterior.

*BBB* is followed by *Flipout* which adds a pseudo-independent perturbation to the weights at each training point $\mathbf{x}_n$ in the mini-batch, namely, $\mathbf{w}_n = \mu + (\sigma \circ \epsilon)R_n$, where $R_n$ is the random sign matrix. Intuitively, the weights get flipped symmetrically around the mean with probability $0.5$.

*Neural Linear Model* (*NLM*) is an alternative to Bayesian NN. It places a prior distribution only on the last layer's weights, and learns point estimates for the remaining layers. One can interpret the output of these layers as a basis defined by the feature embedding of the data. The last layer of *NLM* performs Bayesian linear regression on this feature basis. *NLM* provides tractable inference under the Gaussian assumption on likelihood; we get analytical solution for the posterior distribution.

**Inference through the higher-order evidential distribution**. It is also possible to infer parameters of the posterior predictive directly, using Bayesian hierarchical modeling [6, 7]. In *Deep Evidential Regression* (*DER*), the higher-order, evidential prior is placed over the Gaussian likelihood function. Choosing Normal Inverse-Gamma (NIG) prior yields an analytical solution for the model evidence which is maximized by the optimization objective with respect to the NIG hyperparameters. *DER* also proposes an evidence regularizer which minimizes evidence on incorrect predictions. The posterior predictive mean and variance are computed analytically using the learned hyperparameters.

# 3   Uncertainty quantification in neural differential equations

Instead of learning a deterministic solution $\tilde{u}_N$, we now aim to learn a probabilistic solution $u_\theta$, characterized by a posterior predictive distribution. We estimate it using some probabilistic model $g_\theta$ parametrized by $\theta$.

## 3.1   Proposed approach

UQ methods described in Section 2.2 rely upon the assumption that the likelihood function is Gaussian, centered at the model's prediction and evaluated at the observed data points. Namely, *DER* and *NLM* use it to derive the analytical form of a loss function and a posterior distribution, respectively. *BBB* and *Flipout* can in principle use any likelihood in the loss function, but it has to be of known analytical form. In case of DEs, a natural way of computing likelihood is to evaluate it at the *residual $\mathcal{R}$ on the training domain* $\mathbf{x}_t$, which can be seen as a counterpart to the observed data points in the classical setting. However, we are left with an open problem of choosing the underlying distribution for the likelihood function. It makes sense to assume that the probability density is high enough for values close to zero, but no further assumptions immediately follow. E.g., it may happen that the limitations of NN architecture do not allow for the perfect fit, i.e., the distribution of residuals is not centered around zero. To circumvent this problem, we propose an alternative way of computing likelihood.

In this first work on UQ for neural DE solvers, we will focus on comparing predictions outside of the training domain given by different UQ methods, leaving the detailed treatment of the model fit and its associated uncertainty for future work. Although in Bayesian framework this uncertainty also affects the uncertainty outside of the training domain, we hypothesize that even a simplified treatment, i.e., without using residuals' distribution, gives reasonable uncertainty estimates. We propose a two-stage training procedure:

1) We first train a classical NN on the training domain $\mathbf{x}_t$ to find a *deterministic solution* $\tilde{u}_N$,

2) We use $\tilde{u}_N$ as *observed data* for our probabilistic model $g_\theta$ and define the likelihood using a Gaussian assumption, $p(\tilde{u}_N|\theta) = \prod_{\mathbf{x}_t} \mathcal{N}(\tilde{u}_N; u_\theta, \varepsilon)$.

Now the optimization objective will be trying to align the probabilistic model with the given reference $\tilde{u}_N$ rather than trying to minimize the residuals at all costs. We note that despite interpreting $\tilde{u}_N$ in stage two as observed data rather than a function that solves DE, its associated variance $\varepsilon$ is not of aleatoric nature (i.e. irreducible variance that comes from the noise inherent to the data), as it would be in the classical regression problem. It can be still interpreted as a source of epistemic (reducible) uncertainty coming from the NN model limitations. Here, we consider a simplified treatment of $\varepsilon$. In case of *BBB*, *Flipout*, and *NLM*, we pre-define $\varepsilon$ with some small number. In case of *DER*, we learn $\varepsilon$ along with posterior predictive distribution, but the result is not particularly useful since *DER* does not have direct access to residuals during learning.

Eventually, the probabilistic model allows us to find the posterior predictive distribution of $u_\theta$. In case of *BBB*, *Flipout*, and *NLM*, we have $\tilde{g}_\theta(\mathbf{x}_t, \tilde{u}_N) = u_\theta$, i.e., the model outputs a single instance $u_\theta$. For *BBB* and *Flipout*, the posterior predictive distribution $p(u_\theta|\mathbf{x}_t, \tilde{u}_N)$ is computed as an approximation of integral (2) using sampling; for *NLM*, an analytical form is available. In case of

*DER*, we have $g_\theta(\mathbf{x}_\mathrm{t}, \tilde{u}_N) = (\gamma, \nu, \alpha, \beta)$, where $(\gamma, \nu, \alpha, \beta)$ are the NIG hyperparameters. The mean of the posterior predictive distribution is equal to $\tilde{\gamma}$ and the variance is computed using the remaining hyperparameters. We note that the predictive uncertainty also requires the initial and/or boundary condition enforcement; this way we are able to eliminate unnecessary uncertainties at $\mathbf{x}_\mathrm{c}$.

Main drawbacks of the current approach are the double computational burden and the not so useful uncertainty for NN approximation of the true solution in the traning region; both of them are subject to further improvement.

## 4  Experiments and discussion

We corroborate our theory with experimental results on four equations: 1. Linear ODE for squared exponential, $\frac{du}{dt} = -2tu$; 2. Non-linear ODE for Duffing-type oscillator, $\ddot{u} + \omega^2 u + \epsilon u^3 = 0$; 3. Lotka-Volterra equations (system of non-linear ODEs), $\dot{u} = \alpha u - \beta uv \ \wedge \ \dot{v} = -\delta u + \gamma uv$; 4. Burgers' equation (non-linear PDE), $\frac{\partial u}{\partial t} + u \frac{\partial u}{\partial x} = \nu \frac{\partial^2 u}{\partial x^2}$.

Our implementation is based on a DE solver provided by *neurodiffeq* [4], a Python package built with PyTorch [16]. Since we are considering relatively simple DEs, we use networks with one to three fully-connected hidden layers. For the prior distribution of the weights in *BBB*, *Flipout*, and *NLM*, we use flat Gaussian priors with mean zero. In *BBB* and *Flipout*, we estimate the posterior predictive from 1000 samples. In *DER*, there is no need for choosing a weight prior, but an appropriate regularization parameter has to be chosen instead. Here, we tune the regularization parameter manually.

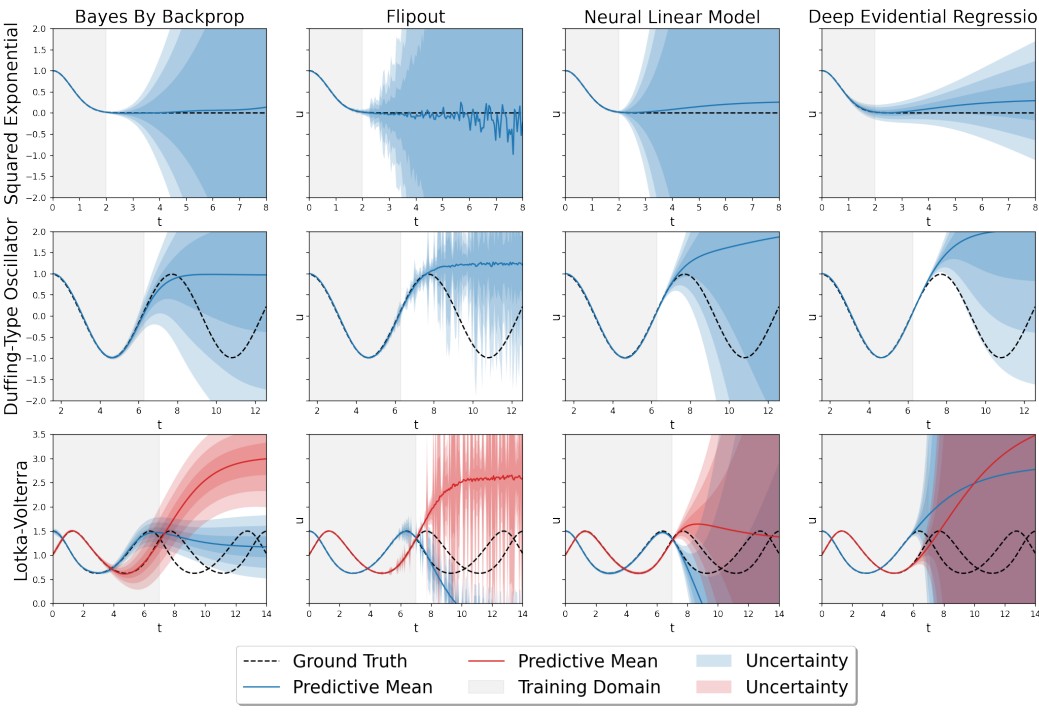

Figure 1: **Uncertainty estimation for neural ODEs**. Probabilistic models give low epistemic uncertainty in the training domain and inflate it outside of the training domain.

We demonstrate the UQ results for ODEs and for PDE in Figure 1 and Figure 2, accordingly. For all ODEs, the deterministic solution is able to approximate the true solution well; we incorporate this fact in our Bayesian inference by choosing small $\varepsilon$ which yields that there is almost no uncertainty in the training domain. We observe that the epistemic uncertainty away from the training domain is high enough for all methods, which is our main desired result in this paper. For Burgers' equation, however, we see that the NN is not able to learn the true solution, and our probabilistic model is underestimating the epistemic uncertainty in the training domain and outside of it. In this case,

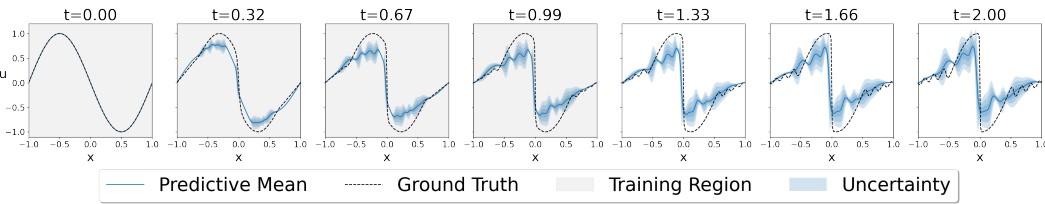

Figure 2: **Uncertainty estimation for NN based solution of Burgers' equation.** We test *Bayes By Backprop* on an example of a non-linear PDE. Uncertainty coming from underfitting is not captured well due to our simplified inference in terms of this type of uncertainty, nevertheless we see that the uncertainty inflates outside of the training domain.

either a better deterministic model or a better UQ methodology is needed. Nevertheless, even for a non-perfect fit, the uncertainty starts inflating outside of the training domain which proves our initial hypothesis.

We have witnessed comparable performance in sampling-dependent (*BBB*, *Flipout*) and sampling-free (*NLM*, *DER*) methods. Given the computational expense of sampling during Bayesian NN training, the latter two methods could be preferable in the case of complex DEs on a multidimensional domain.

We believe that further enhancement in terms of diversifying experiments (e.g., considering more complex high-dimensional DEs) and developing theory (e.g., calibrating $\varepsilon$ with residuals at each optimization step) will help the deep learning based DE solutions to outperform classical ones and lead to their increased presence in applications.

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
