# OpenReview forum: "Uncertainty Quantification in Neural Differential Equations"
_NeurIPS.cc/2021/Workshop/DLDE — DLDE Workshop -- NeurIPS 2021 Poster_

### Official Review · Reviewer_s2wG · 2021-10-11
**Interesting adaptation of uncertainty quantification methods for neural differential equation solvers, but with questionable applicability to practical problems**

**Confidence:** 4

**Review:**

The authors propose an adaptation of popular Bayesian uncertainty-quantification methods for neural network-based differential equation solvers. The problem is of interest to the community and the proposed method is straightforward but interesting. The reviewer has the following concerns which are mainly centered around the scalability of the approach:

1. The proposed approach requires the model to be trained twice (lines 97-99), which is computationally expensive and significantly limits the applicability of this method on real-world use cases involving complex high dimensional ODEs/PDEs and a large number of time-points.

2. Fixing the epistemic uncertainty hyperparameter $\varepsilon$ (lines 98-99) seems quite ad-hoc. While one set $\varepsilon$ to be very small for the kind of simple low dimensional ODEs used in the experiments (as the neural network can easily approximate the solution), choosing an appropriate value of $\varepsilon$ becomes a nontrivial hyperparameter selection task for complex ODEs where learning a good approximation becomes challenging for the NN. Such a scenario is demonstrated within the paper itself, as the NN is unable to fit the solution to Burger's equation (lines 136-138). The task of choosing $\varepsilon$ adds a hyperparameter selection step to the already expensive two-stage training procedure, thereby further limiting scalability to a significant extent.

3. Considering the fact that the model ensures that the initial/boundary conditions are always satisfied (lines 44-47), the uncertainty blow-up observed near t=0 for the Neural Linear Model and Deep Evidential Regression experiments (Figure 1) is quite concerning, and requires further explanation.

4. The paper considers only low dimensional and simple ODE/PDE examples for experimental evaluation. Applications where uncertainty quantification is of importance generally involve complex high-dimensional differential equations. It would be interesting to see experiments on low dimensional chaotic dynamical systems such as the Lorenz Attractor, as well as more complex ODEs/PDEs with greater practical significance such as the Navier-Stokes equation.

5. Benchmarking against other methods for uncertainty quantification (such as model ensembles or MC Dropout) is absent.

6. The choice of fixing the posterior-predictive distribution to be a Gaussian seems non-standard. To the best of my understanding, the existing literature on Bayesian NNs generally uses Gaussian priors on the weight space. Further explanation motivating this choice would be greatly appreciated.

**Score:**

2: Borderline paper

---

### Official Review · Reviewer_XnZk · 2021-10-11
**Work in uncertainty quantification for NDE solvers would benefit from further analysis.**

**Confidence:** 4

**Review:**

The paper aims to extend some uncertainty quantification method to NN DE solvers.
The paper would benefit from more concrete explanations of: the problem setting (line 46, A should be defined, is there a citation you can add to support lines 43-44?), implications of the assumptions (98-99: why constraining the solution in this form is ok for this problem?), the results (lines 107-108 are unclear as well as the results from Fig 1.), and the corresponding conclusions. The papers concludes that "Given the computational expense of sampling during Bayesian NN training, the latter two methods could be preferable in case of complex DEs on a multidimentional domain." It is unclear why this claim follows from their experiments.




**Score:**

2: Borderline paper

---

### Official Review · Reviewer_SeBV · 2021-10-11

**Confidence:** 3

**Review:**

The authors propose examining the certainty of neural network predictions, for a variety of neural networks that are aimed at solving differential equations.

While the idea and concept is indeed interesting and important, there are quite some fundamental problems that need to be addressed:

-Only low dimensional ODEs and PDEs are considered, and under restrictive conditions (e.g. the solution is constrained to some form of a Gaussian distributions). It is suggested that the authors elaborate on those choices and add experiments that involve more complex PDEs, specifically in higher dimensions.

-No comparison with other methods is provided.

**Score:**

2: Borderline paper

---

### Decision · Program_Chairs · 2021-10-17

**Decision:**

Accept (Poster)

**Comment:**

This paper considers the important problem of uncertainty quantification for differential equation solvers. The work has been selected as a poster for the workshop. The authors may consider this opportunity to address comments made by reviewers.